# Wastewater surveillance in smaller college communities may aid future public health initiatives

Laura Lee[1], Lescia Valmond[2], John Thomas[2], Audrey Kim[2], Paul Austin[1], Michael Foster[1], John Matthews[3], Paul Kim[2], Jamie Newman[1]*

1 School of Biological Sciences, Louisiana Tech University, Ruston, Louisiana, United States of America,
2 Department of Biology, Grambling State University, Grambling, Louisiana, United States of America,
3 Trenchless Technology Center, Louisiana Tech University, Ruston, LA, United States of America

* jjnewman@latech.edu

**Data Availability Statement:** All relevant data are within the paper and its Supporting Information files.

## Abstract

To date, the COVID-19 pandemic has resulted in over 570 million cases and over 6 million deaths worldwide. Predominant clinical testing methods, though invaluable, may create an inaccurate depiction of COVID-19 prevalence due to inadequate access, testing, or most recently under-reporting because of at-home testing. These concerns have created a need for unbiased, community-level surveillance. Wastewater-based epidemiology has been used for previous public health threats, and more recently has been established as a complementary method of SARS-CoV-2 surveillance. Here we describe the application of wastewater surveillance for SARS-CoV-2 in two university campus communities located in rural Lincoln Parish, Louisiana. This cost-effective approach is especially well suited to rural areas where limited access to testing may worsen the spread of COVID-19 and quickly exhaust the capacity of local healthcare systems. Our work demonstrates that local universities can leverage scientific resources to advance public health equity in rural areas and enhance their community involvement.

## Introduction

Since the COVID-19 pandemic was declared, there have been over 570 million infections and over 6 million deaths worldwide [1]. Over the past two years, mutations during viral replication coupled with the unchecked global spread of COVID-19 have led to the emergence of more transmissible variants of concern. The first of these variants, the novel SARS-CoV-2 B.1.617.2 (Delta), was identified in India in December 2020 [2]. This variant was the catalyst for a COVID-19 surge seen in July 2020 [3]. Similarly, the novel SARS-CoV-2 B.1.1.529 (Omicron) variant emerged in November 2021 and resulted in yet another surge and a record number of cases across the United States [4].

Rapid diagnostic testing is a critical tool for breaking viral transmission chains and provides data on the prevalence and spread of infectious diseases that can inform public health decision making. However, in the case of COVID-19, each surge was exacerbated by limited supply and

**Funding:** JJN, PK Rockefeller Foundation Regional Accelerator PK received an. Institutional Development Award (IDeA) from the National Institute of General Medical Sciences of the National Institutes of Health under grant number P2O GM103424-20 The funders had no role in study design, data collection and analysis, decision to publish, or preparation of the manuscript.

**Competing interests:** The authors have declared that no competing interests exist.

access to testing in the US, meaning that often the reports were underestimating the number of infected individuals. More at-home testing and more mild or asymptomatic cases due to acquired immunity have further widened the discrepancy between caseload reporting and actual infections [5]. All of this then points to a need for additional community surveillance of SARS-CoV-2.

Wastewater-based epidemiology (WBE), used for decades to monitor chemicals and pathogens through the analysis of sewage, has been propelled into the spotlight during the pandemic as a complementary tool for estimating COVID-19 prevalence in a community [6–9]. Compared to large-scale diagnostic testing programs, WBE avoids bias, is non-invasive, and is less constrained by limited testing capacity [10]. Although the conversion of viral RNA copy number in sewage to infected individuals is complicated by biological and sewershed variability [11], WBE can still capture near-real-time longitudinal trends. Importantly, WBE has been shown to predict case surges by approximately 5–14 days, providing opportunities for public health and epidemiologic intervention [12, 13].

Rural areas that have fewer resources than urban areas have lagged in testing rates while also being home to a more vulnerable population [14, 15]. In a low testing environment, a WBE approach is especially useful as it indicates infection levels and encourages allocation of resources to those communities to prevent or at least minimize the impact of an outbreak. Previously, WBE has been implemented on various campuses where sufficient clinical testing is not feasible, providing a basis for SARS-CoV-2 monitoring and outbreak mitigation [16–19]. Here we report on the analysis of longitudinal samples collected throughout the Delta and Omicron surges in rural Lincoln Parish, Louisiana. We assess the effect of fecal normalization and compare temporal trends of SARS-CoV-2 in the wastewater to confirmed cases to estimate the sensitivity of wastewater surveillance. To our knowledge, this is the first study of its kind where two public, rural primarily undergraduate campuses leveraged limited resources to forge partnerships and produce data to inform public health.

## Methods

Wastewater from the city of Ruston was collected and analyzed at Louisiana Tech University (LTU) and wastewater from the city of Grambling and the Grambling State University campus was collected and analyzed at Grambling State University (GSU). The same protocol was followed by the two laboratories whenever possible with any differences described below.

### Wastewater sample collection

**City of Ruston wastewater.** Wastewater samples were collected from the single wastewater treatment facility in Ruston, Louisiana, the Ruston Water Treatment Plant. A total of 2.4 L of wastewater was collected with a refrigerated autosampler over 12 hours from 7:00 am to 7:00 pm on the day of sampling. From that, a 100 mL composite sample (2 50 mL tubes) was collected, and heat inactivated in a water bath at 60°C for 90 minutes with one turn at 45 minutes. This was done to adhere to safety protocols required for bringing wastewater back to the university laboratory space. Literature indicates that this is not required as the virus viability is greatly reduced in stool samples [20]. Following inactivation, these samples were stored at 4°C to be picked up that same week. Composite samples were collected from 5/26/2021 to 5/4/2022 and processed at LTU.

**City of Grambling and GSU campus wastewater.** The lift stations most proximal to the City of Grambling Wastewater Treatment Plant that convey wastewater from the city sewershed (32.516403, −92.717004) and GSU campus sewershed (32.515078, −92.718722) were selected for weekly sampling. Composite samples (150 mL per hour for 24 hours) were

collected on ice from Tuesday morning to Wednesday morning each week from 4/27/2021 to 5/3/2022 and immediately processed at GSU.

## Wastewater sample processing

**City of Ruston wastewater.** The heat-inactivated wastewater samples were centrifuged at 4696 × g for 30 minutes to remove debris. The 2 50 mL samples were combined into one 20 mL aliquot of supernatant, and viral matter was precipitated using 10% polyethylene glycol (PEG) and 2.25% NaCl with gentle inversion until reagents dissolved based on the method described by Hebert [21]. This solution was stored at 4˚C until centrifugation at 12,000 × g for 120 minutes at 4˚C. The resulting pellet was resuspended in 140 μL nuclease-free water and stored at −80˚C until RNA extraction could be completed.

**City of Grambling and GSU campus wastewater.** Viruses were concentrated from 60 mL of clarified supernatant via PEG/NaCl precipitation as at LTU. The wastewater/PEG/NaCl solution was centrifuged at 12,000 × g for 99 minutes at 4˚C. The resulting pellet was resuspended in 140 μL of PBS for immediate extraction.

## Viral RNA extraction

**City of Ruston wastewater.** Viral RNA was extracted from 140 μL of resuspended samples using the QIAGEN QIAmp MiniKit (#52904) according to the manufacturer's protocol. The extracted RNA was eluted from the column using 40 μL elution buffer. The RNA purity was determined using the BioTek Cytation Take5 plate reader with A260/A280 ratios averaging 1.7–2.2.

**City of Grambling and GSU campus wastewater.** Viral RNA was extracted as at LTU with one modification. Carrier RNA (5.6 μg) was added to the lysis buffer of all samples except the initial sample to allow for quality assessment of the extracted RNA. RNA integrity measured using the Invitrogen Qubit RNA IQ Assay indicated 61% large or structured RNA and 39% small RNA.

## RT-qPCR and fecal normalization

**City of Ruston wastewater.** 10 μL of viral RNA was used in a 20 μL reaction to create cDNA using the Applied Biosystems High-Capacity Reverse Transcription Kit with RNAse Inhibitor (#4368814) according to the manufacturer's protocol. The resulting cDNA was stored at −20˚C for up to a week prior to quantification of SARS-CoV-2 RNA presence. SARS-CoV-2 presence was measured via qPCR detection using the IDT 2019-nCoV RUO Kit (#10006713) containing the CDC 2019-nCoV diagnostic primer/probe mixes for the N1 and N2 gene targets (IDT #10006625) with the TaqMan Universal PCR Master Mix (Thermo Fisher #4304437) according to the manufacturer's protocol. Each reaction contained 10 μL master mix, 1.5 μL primer/probe mix for N1 or N2, 2 μL target sample or no-template control, and was brought to a total volume of 20 μL using nuclease-free water. Reactions were run at 95˚C for 10 minutes, 50 cycles of 95˚C for 15 seconds followed by 60˚C for 1 minute. For qPCR detection of PMMoV, a primer/probe mix previously described by Haramoto et al. was used instead of the N1 or N2 primer/probe mixes [22]. Amplification parameters were 25˚C for 10 minutes, 95˚C for 3 minutes, 45 cycles of 95˚C for 15 seconds followed by 60˚C for 1 minute. All qPCR reactions were done in triplicate. The reactions were prepared in an Applied Biosystems MicroAmp Fast 96 well reaction plate (#4346906) sealed with MicroAmp clear optical adhesive film (#4311971) and analyzed on an Applied Biosystems StepOnePlus RT-qPCR machine. N1 and N2 samples were quantified using a serial dilution for each gene target. IDT 2019-nCoV N positive control plasmid (#10006625) was used at concentrations ranging

from $4 \times 10^5$ to $4 \times 10^1$ copies per reaction. PMMoV samples were quantified using a 68 bp DNA oligo containing the target region in a serial dilution ranging from $2.4 \times 10^7$ to $2.4 \times 10^1$ copies per reaction [21]. For fecal normalization, the genome copies or GC/mL of N1 and N2 were divided by the GC/mL of PMMoV to obtain a unitless ratio of SARS-CoV-2 to PMMoV [23]. Molecular-grade water was used as a method blank to validate protocol and technique. No amplification was observed in the method blank or in the no-template controls. The limit of detection defined as 95% probability of detection was estimated using logistic regression at 11.48 GC/reaction. Non-detect replicates were excluded and only quantifiable replicates were used in subsequent analysis.

**City of Grambling and GSU campus wastewater.** Reverse transcription was performed as at LTU. The cDNA was stored at −20°C for 1 to 3 days prior to analysis for SARS-CoV-2 and 1 to 8 weeks prior to analysis for PMMoV. SARS-CoV-2 and PMMoV were quantified as at LTU using the IDT 2019-nCoV RUO kit but with the IDT PrimeTime Gene Expression Master Mix (#1055772) according to the manufacturers' protocols. All qPCR reactions were assembled in triplicate. Samples were prepared in a Bio-Rad HSP9601 clear well plate sealed with a Bio-Rad MSB1001 adhesive optical film and analyzed on a Bio-Rad CFX Connect instrument. Quantification cycle (Cq) was determined using Bio-Rad CFX Manager 3.1. N1 and N2 in the samples were quantified using serial dilutions of two standards: (1) the IDT 2019-nCoV_N positive control plasmid containing the complete nucleocapsid gene at concentrations ranging from $2 \times 10^4$ to $2 \times 10^1$ plasmid copies per reaction and (2) the ATCC VR3276SD synthetic RNA, reverse transcribed following the same protocol as sample RNA, at concentrations ranging from $2 \times 10^4$ to $2 \times 10^1$ RNA copies input to reverse transcription (Fig 1). PMMoV quantification and normalization was performed as described at LTU.

**Pepper mild mottle virus optimization for copy number normalization.** Plasmid standards have been reported to overestimate the viral load due to inefficient amplification of the supercoiled template during earlier cycles. In our study, the synthetic RNA-derived standards were slightly less efficient (N1 93.7%, N2 90.1%) thus the positive control plasmid was used for quantification of gene copies (Fig 1). PMMoV in the samples was quantified using serial dilutions of a 68 bp DNA oligo containing the target region at concentrations ranging from 2.4 x 107 to 2.4 x 101 copies per reaction.

The N1 primer set and the positive control plasmid generated a standard with an R2 value of 0.978 and 97.8% efficiency (slope −3.39, intercept 42.71). The N2 primer set and the positive

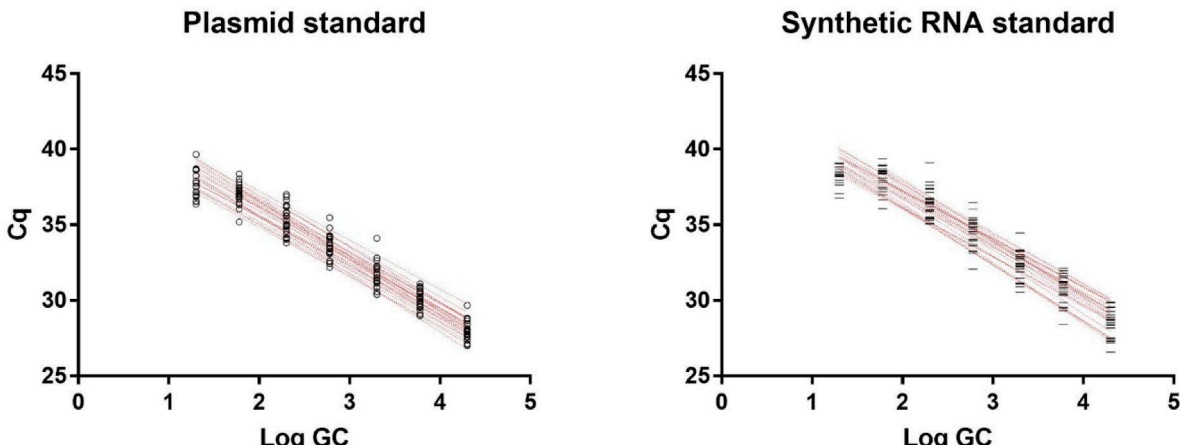

**Fig 1.** Standard curves obtained from IDT 2019-nCoV_N positive control plasmid (left) and reverse-transcribed ATCC VR3276SD synthetic RNA (right).

control plasmid generated a standard with an R2 value of 0.991 and 95.9% efficiency (slope −3.46, intercept 43.549). The PMMoV primer set generated a standard with an R2 value of 0.996 and 88.7% efficiency (slope −3.65, intercept 42.15).

The presence of PCR inhibitors in the sample matrix was tested by spiking 2 x 105 copies of the IDT 2019-nCoV_N positive control plasmid into extracted wastewater in which no SARS-CoV-2 was detected and comparing the Cq to the same concentration of plasmid in the standard. There was no significant difference in Cq between the wastewater extract (24.87) and standard (24.76) suggesting no PCR inhibition.

## Results

Ruston, a city in rural Lincoln Parish, Louisiana with a population of approximately 22,000 people, is home to Louisiana Tech University, a public university with an enrollment of approximately 12,000 students. The city of Ruston has a single wastewater treatment facility that services wastewater for over 90% of the population. To carry out wastewater surveillance of SARS-CoV-2, we collaborated with the city's wastewater treatment facility to obtain samples for analysis. Samples were not able to be collected every week due to inclement weather, critical mechanical difficulties at the treatment facility, or absence of staff (S1 Table). The city of Grambling, also in Lincoln Parish, has a population of 5,150 residents as of the July 8, 2021, census. Grambling State University (GSU) has an enrollment of 5,438 students with 2,005 students living on campus and 226 faculty and 367 staff members working on campus during the Fall 2021 academic term, and 1,818 students, 197 faculty, and 374 staff members on campus during the Spring 2022 academic term.

We determined the concentration (genome copies or GC/L) of pepper mild mottle virus (PMMoV), a fecal indicator that is frequently used to normalize wastewater testing and account for fluctuations in population or precipitation during the collection period (Fig 2). PMMoV is highly abundant in raw wastewater with concentrations ranging from $10^5$ to $10^9$ GC/L typically being reported in the literature [24]. We detected PMMoV in all samples from all sites with average concentrations in the order of $10^8$ GC/L in Ruston and $10^6$ GC/L in the smaller Grambling community. In Grambling, the PMMoV concentrations in the city

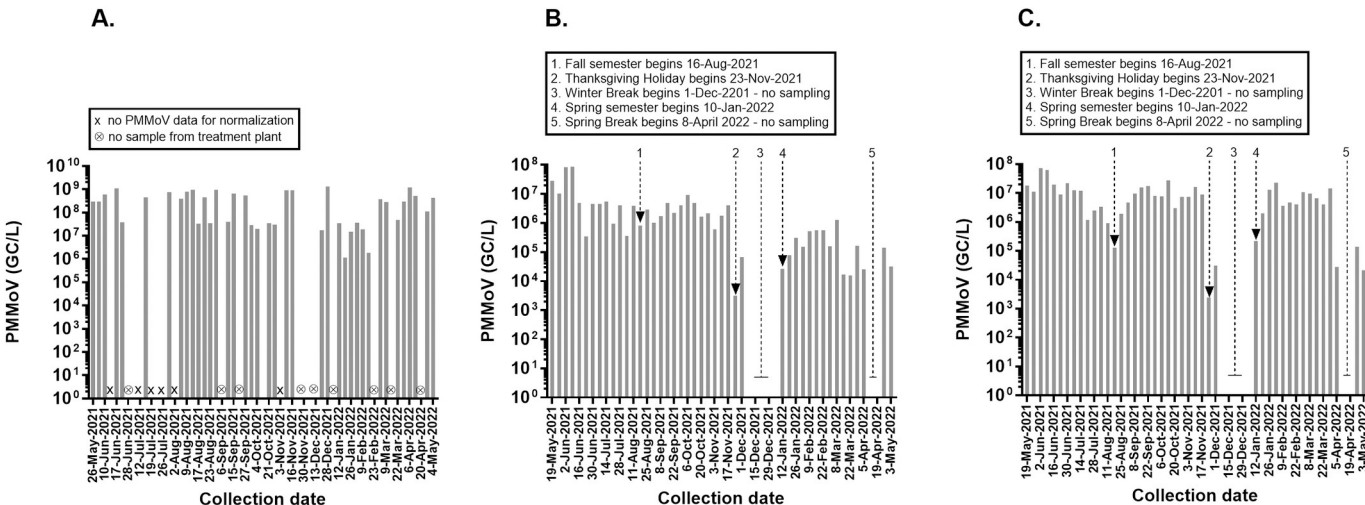

**Fig 2. Pepper mild mottle virus (PMMoV) in wastewater.** PMMoV concentrations expressed as genome copies or GC/L in the wastewater of Ruston (A), Grambling (B), and Grambling State University (C). The timeline is annotated with key events and dates including dates when no wastewater samples were collected or PMMoV amplification failed. For full table of reporting in Ruston see S1 Table.

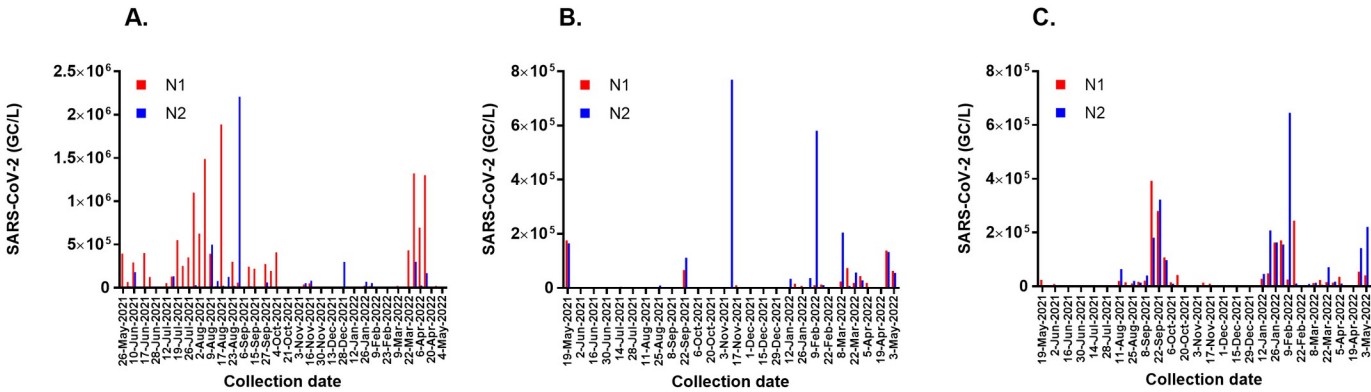

**Fig 3. SARS-CoV-2 in wastewater.** N1 and N2 concentrations expressed as genome copies or GC/L in the wastewater of Ruston (A), Grambling (B), and Grambling State University (C).

mirrored the GSU campus, which in turn were highly dependent on the academic calendar with a high of $2.7 \times 10^7$ GC/L detected during Homecoming week and a low of $2.4 \times 10^3$ GC/L detected during the Thanksgiving Break. PMMoV concentrations in Ruston were not as coupled to the LTU academic calendar and various events in the community that brought people to Ruston, LA may account for spikes in the wastewater signal.

The non-normalized wastewater concentrations of SARS-CoV-2 in Ruston, Grambling, and GSU are expressed as GC/L in Fig 3. In Ruston, N1 or N2 genes were detected in 45 of 46 wastewater samples with values ranging from $1 \times 10^3$ GC/L to $1.1 \times 10^6$ GC/L. In January and February 2022, the concentrations of SARS-CoV-2 detected were unexpectedly low considering this was during the peak of the Omicron surge in Louisiana. In Grambling, N1 or N2 was only detected in 19 of 51 wastewater samples and at much lower concentrations than in Ruston, often only exceeding the limit of detection when viral loads were relatively high in the GSU campus sewershed. On the GSU campus, we observed two spikes in the wastewater signal associated with the Delta and Omicron surges against a low baseline signal in 29 of 51 wastewater samples.

The PMMoV-normalized wastewater concentrations of SARS-CoV-2 expressed as unitless ratios are presented with city caseload data (Fig 4). Normalizing for fecal load reveals that SARS-CoV-2 concentrations in Ruston wastewater during the Omicron surge were

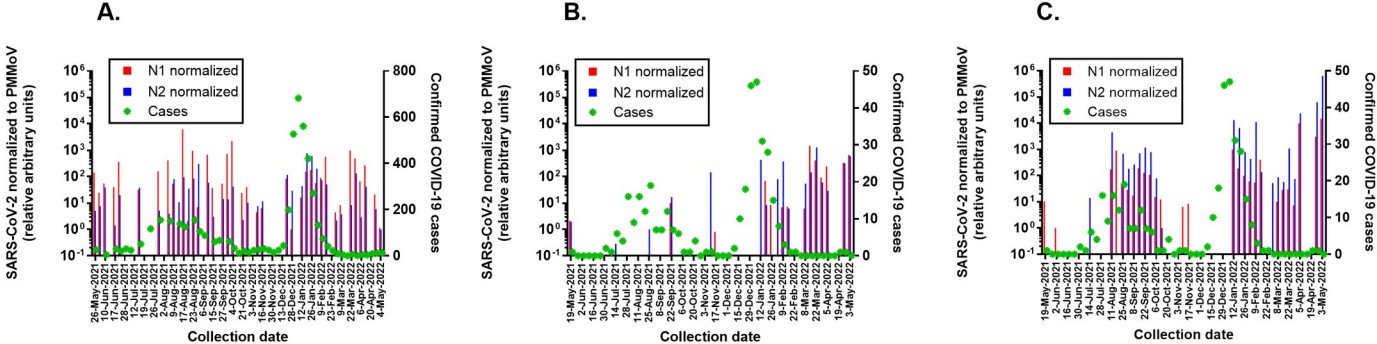

**Fig 4. PMMoV-normalized SARS-CoV-2 in wastewater and weekly caseloads.** N1 and N2 concentrations in the wastewater of Ruston (A), Grambling (B), and Grambling State University (C) were divided by PMMoV concentrations to obtain a unitless ratio that normalizes for fecal load. All ratios are relative to the lowest ratio set arbitrarily as $10^0$ and plotted on the left Y axis. Total weekly caseloads in zip codes 71270 and 71273 (Ruston) and zip code 71245 (Grambling/Grambling State University) are plotted on the right Y axis.

comparable to those detected during the Delta surge. The low GC/L observed in the non-normalized data may have been due to viral losses in the sewage system. In the Grambling community, there was little correlation between normalized SARS-CoV-2 concentration in city wastewater and confirmed infections in the city as both appear to be primarily driven by the influx and efflux of people on the GSU campus. Because SARS-CoV-2 can be shed in feces early in the course of COVID-19 infection, it has been proposed that wastewater surveillance can serve as an early warning system [25–27]. On the GSU campus however, the wastewater signal appeared to lag or at best coincide with the increase in confirmed infections during the Delta surge. The sudden influx of thousands of students at the beginning of the academic year, all of whom were screened if moving into campus housing, precludes using wastewater surveillance as a forecasting tool in this instance. There was no data collected from the campus sewershed during Winter Break, but it is reasonable to expect a similar lack of predictive power during the initial Omicron surge which coincided with the return of students to campus. Other limitations of our study include the lack of a matrix control to assess viral recovery and the lack of normalization for daily wastewater flow.

## Conclusion

Here we demonstrate the ability of smaller universities to serve as public health resources in their community by engaging undergraduate students in wastewater surveillance. Monitoring of wastewater for SARS-CoV-2 is especially critical as we enter a time of at-home testing and generally less official reporting. This is a trend confirmed by data from The Institute for Health Metrics and Evaluation, our analysis of which suggests that during the peak of the Delta surge, approximately 41% of cases are estimated to have been reported, compared to roughly 21% during the peak of the Omicron surge, and less than 10% toward the end of this study period [28]. Incorporation of wastewater surveillance at college campuses, as seen at GSU, is crucial as university cases have been shown to influence the case numbers and characteristics of the broader community in which they are located [29]. Ruston, Grambling, and GSU all saw an increase in viral wastewater concentrations in January and February 2022, corresponding to increased regional and national caseloads. However, in looking at the data (Fig 4), the same relative genome copies corresponded to higher local caseloads in earlier months than what was being reported in April and May 2022, suggesting more recent under-reporting of COVID-19 cases in communities. This is a significant problem because under-reporting may lead to a false sense of security among the public and hinder data-driven decisions by policymakers. Overall, this study indicates the need to continue regular surveillance and heed the warnings of viral genome concentrations in the wastewater as a representative indicator of community health as well as COVID-19 cases in the area [6, 30–33]. Smaller communities do not always have access to the same resources or information available in larger cities. In these cases, it is critical that the university community become engaged in monitoring and supporting public health initiatives. Integration of facilities designed for wastewater surveillance in low income and rural communities is possible and could enhance understanding of public health [34]. The ability for two campuses to initiate this type of surveillance and train undergraduate students to be a part of the research programs establishes a model for this type of work going forward that will allow universities to participate in public health.

## Supporting information

**S1 Table. Description of collection and/or detection failure in Ruston, LA WBE.**
(TIF)

## Acknowledgments

We would like to thank Helaina Desentz, Louisiana Department of Health, Region 8 Epidemiologist for providing local caseload data and Casey Jackson for wastewater sampling through the duration of this project.

## Author Contributions

**Conceptualization:** John Matthews, Paul Kim, Jamie Newman.

**Data curation:** Laura Lee.

**Formal analysis:** Laura Lee, Paul Kim, Jamie Newman.

**Funding acquisition:** Paul Kim, Jamie Newman.

**Investigation:** Lescia Valmond, John Thomas, Audrey Kim, Paul Austin, Michael Foster, Paul Kim, Jamie Newman.

**Methodology:** Laura Lee, Paul Austin.

**Project administration:** Jamie Newman.

**Supervision:** Audrey Kim, Paul Kim, Jamie Newman.

**Validation:** Audrey Kim.

**Visualization:** Paul Kim.

**Writing – original draft:** Laura Lee, Jamie Newman.

**Writing – review & editing:** Laura Lee, Paul Kim, Jamie Newman.

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
