## [Decision Letter · Decision Letter 0]

26 Jul 2022

PONE-D-22-16566Wastewater surveillance in smaller college communities may aid future public health initiatives

PLOS ONE

Dear Dr. Newman,

Thank you for submitting your manuscript to PLOS ONE. After careful consideration, we feel that it has merit but does not fully meet PLOS ONE’s publication criteria as it currently stands. Therefore, we invite you to submit a revised version of the manuscript that addresses the points raised during the review process.

Please provide more detailed information on the methodology used and QA/QC protocols in your revised manuscript. Also please cite recent published papers on this topic. 

We look forward to receiving your revised manuscript.

Kind regards,

Asli Aslan

Academic Editor

PLOS ONE

Journal Requirements:

Reviewers' comments:

Reviewer's Responses to Questions

**Comments to the Author**

1. Is the manuscript technically sound, and do the data support the conclusions?

Reviewer #1: Partly

Reviewer #2: Yes

2. Has the statistical analysis been performed appropriately and rigorously? 

Reviewer #1: N/A

Reviewer #2: Yes

3. Have the authors made all data underlying the findings in their manuscript fully available?

Reviewer #1: No

Reviewer #2: Yes

4. Is the manuscript presented in an intelligible fashion and written in standard English?

Reviewer #1: Yes

Reviewer #2: Yes

5. Review Comments to the Author

Reviewer #1: The authors of the manuscript PONE-D-22-16566 have conducted a nice study but there are some loopholes in the manuscript which needs to be fixed before accepted for publication. The points to be addressed are mentioned below;

Please reframe the 1st sentence of the abstract. The part “since its declaration” doesn’t sound good. Maybe the sentence can be reframed to “Since the declaration of COVID-19 pandemic” or any other of your choice.

In the introduction, please cite references where WBE has been imposed at school/ college/ university level. Few of the references are provided below;

https://doi.org/10.1371/journal.pone.0270168

https://doi.org/10.1016/j.scitotenv.2021.146408

https://doi.org/10.1016/j.scitotenv.2021.146749

Please highlight the novelty of the study in the last paragraph of the introduction.

Why was the ww sample from one WWTP heat inactivated but other one not?

Was there any field blank, method blank, extraction blank?

What about negative controls?

What are the QA/QC measures taken into consideration?

Was the LoD, LoQ determined?

Was any RT-qPCR inhibition noticed?

Incorporate statistical analysis.

Critical scientific explanation for the obtained results are missing.

Please provide better picture resolution of the figures and the fonts should be legible.

Please incorporate references in support of the obtained results. The literatures provided below when cited would enhance the manuscript;

https://doi.org/10.1371/journal.pone.0266407

https://doi.org/10.1128/aem.01740-21

https://doi.org/10.1016/j.scitotenv.2021.152503

https://doi.org/10.1016/j.scitotenv.2021.150264

https://doi.org/10.1371/journal.pwat.0000007

https://doi.org/10.1021/acsestwater.2c00052

Reviewer #2: This is a well-written report of SARS-CoV-2 wastewater monitoring conducted at three rural locations. The analysis is clearly reported and follows one conventional viral concentration procedure that has been commonly used despite its analytical challenges and drawbacks (e.g., <0.51% recovery of phi6 used as an indicator for SARS-CoV-2 in this paper: https://www.sciencedirect.com/science/article/pii/S0048969721058009 )

My primary concern is that the PEG precipitation method does not necessarily appear to generate concentrations that provide reliable detection or trend information. For example, the authors report low concentrations in the wastewater during the Omicron surge, lack of correlation with observed clinical case trends, and lag of wastewater data (rather than lead over clinical cases). Some of these observations could be explained by poor clinical testing or other environmental factors, but it’s also possible some of the observations result from analytical issues. The analysis appears to have been performed carefully and consistently, so I wonder how effective this particular method actually is at low concentrations (which may be most relevant in these communities). Generally, we’ve found the PEG precipitation method to perform poorly. The authors should provide more context in terms of the success (or not) of the PEG method applied in generating reliable results that can be used to capture trends in infection rates for similar contexts. If other studies have demonstrated the particular utility of this method, this would build confidence in the wastewater results reported. The normalization with PMMoV also does not appear to improve the correlations with clinical cases—is this consistent with what others have seen using this method in similar contexts? Were correlations between wastewater and clinical cases in this study better or worse with and without PMMoV normalization?

I absolutely agree that rural areas and small communities can benefit from wastewater monitoring, but more sensitive and reliable methods may be needed to confidently report results (e.g., analysis of primary clarifier sludge is much more sensitive as an overall method for detection https://pubs.rsc.org/en/content/articlehtml/2022/ew/d1ew00826a ). If the wastewater measurements don’t consistently lead over clinical cases, the tool is not useful for early warning. Only analytical results (non-normalized) from WWTP plant (Figure 2C) appears to have consistent detection for N1 and N2 as well as trends that fit with expectations from the known surges in infections. Can the authors explain the poor agreement between N1 and N2 in Figure 2a?

I recommend that the authors report the full suite of Minimum Information for Publication of Quantitative Real-Time PCR Experiments (MIQE) to improve the completeness of the reported results ( for further information, see Bivins et al, 2021 https://www.ncbi.nlm.nih.gov/pmc/articles/PMC8341816/ ).

Especially given higher rates of non-detects, the limit of detection (LOD) in terms of genome copies / ul extract as well as genome copies/ ml wastewater sample. Standard curves are shown, but I don’t believe the LOD was reported. How non-detects were treated in triplicate qPCR analysis should also be reported (e.g., if 1 or more results of the triplicate were below the LOD, were the replicates assigned a value such as the LOD, were the results censored, or how else where they handled?).

6. PLOS authors have the option to publish the peer review history of their article (what does this mean?). If published, this will include your full peer review and any attached files.

Reviewer #1: No

Reviewer #2: No

---

## [Author Response · Author response to Decision Letter 0]

6 Aug 2022

Style requirements have been reviewed and followed

Data has been uploaded as Supporting Information files.

Data has been uploaded as Supporting Information files at the time of resubmission and reflected in the cover letter.

Captions have been added as is appropriate to Supporting Information files.

Reviewers' comments:

Reviewer's Responses to Questions

Comments to the Author

1. Is the manuscript technically sound, and do the data support the conclusions?

Reviewer #1: Partly

Reviewer #2: Yes

2. Has the statistical analysis been performed appropriately and rigorously? 

Reviewer #1: N/A

Reviewer #2: Yes

3. Have the authors made all data underlying the findings in their manuscript fully available?

Reviewer #1: No

Reviewer #2: Yes

4. Is the manuscript presented in an intelligible fashion and written in standard English?

Reviewer #1: Yes

Reviewer #2: Yes

5. Review Comments to the Author

Reviewer #1: The authors of the manuscript PONE-D-22-16566 have conducted a nice study but there are some loopholes in the manuscript which needs to be fixed before accepted for publication. The points to be addressed are mentioned below;

Please reframe the 1st sentence of the abstract. The part “since its declaration” doesn’t sound good. Maybe the sentence can be reframed to “Since the declaration of COVID-19 pandemic” or any other of your choice.

In the introduction, please cite references where WBE has been imposed at school/ college/ university level. Few of the references are provided below;

https://doi.org/10.1371/journal.pone.0270168

https://doi.org/10.1016/j.scitotenv.2021.146408

https://doi.org/10.1016/j.scitotenv.2021.146749

We appreciate the suggestions for additional reference of WBE in academic institutions and the suggested citations. The abstract has been edited and relevant citations added to the introduction.

Please highlight the novelty of the study in the last paragraph of the introduction.

The introduction has been modified to reflect the fact that, to our knowledge, this was the first study done across two rural, primarily undergraduate university campuses that leveraged limited resources to forge partnerships and produce data that may inform public health in the future.

Why was the ww sample from one WWTP heat inactivated but other one not?

Many labs do not heat inactive wastewater samples and heat inactivation has been shown to have no significant impact on viral detection. As the studies reported in our manuscript were performed on two different campuses, with different safety protocols, the wastewater for one site was heat treated at the treatment facility before being taken to the university research lab in order to adhere to university safety protocol. Pecson BM et al. Environ Sci.: Water Res. Technol. 2021

Was there any field blank, method blank, extraction blank?

Molecular grade water was used in the initial extraction reactions and resulted in no amplification. To conserve resources, this was not repeated with each reaction once confirmation of protocol was obtained. This has been noted in the methods.

What about negative controls?

No template controls were used as negative controls. This has been noted in the methods section.

What are the QA/QC measures taken into consideration?

As described in methods, RNA quality was first assessed in the absence of carrier RNA using an RNA IQ assay and identified 61% long, structured RNA. We also measured A260/A280 ratios in the range of 1.7 to 2.2, indicating acceptable purity of the extracted RNA, and this has been added to the methods.

Was the LoD, LoQ determined?

The LOD was determined using logistic regression and the method has been updated.

Was any RT-qPCR inhibition noticed?

No inhibition was noticed. This was previously mentioned in the supplemental information and has been moved to the methods section for clarification.

Incorporate statistical analysis.

We have elaborated on analysis, quality control, and technique validation in our methods section.

Critical scientific explanation for the obtained results are missing.

We are not sure what the reviewer is asking for with this comment. We have attempted to clearly explain all of our data, provide literature references to support our findings and conclusions, and addressed all other comments to enhance the manuscript for publication.

Please provide better picture resolution of the figures and the fonts should be legible.

The figures were exported at high resolution (600 dpi) and in the upload of files it was noted they will not appear high resolution in the pdf compiled for review. This should be addressed in the final proof of the paper.

Please incorporate references in support of the obtained results. The literatures provided below when cited would enhance the manuscript;

https://doi.org/10.1371/journal.pone.0266407

https://doi.org/10.1128/aem.01740-21

https://doi.org/10.1016/j.scitotenv.2021.152503

https://doi.org/10.1016/j.scitotenv.2021.150264

https://doi.org/10.1371/journal.pwat.0000007

https://doi.org/10.1021/acsestwater.2c00052

We appreciate this suggestion. Additional references have been added to support the results of the study. 

Reviewer #2: This is a well-written report of SARS-CoV-2 wastewater monitoring conducted at three rural locations. The analysis is clearly reported and follows one conventional viral concentration procedure that has been commonly used despite its analytical challenges and drawbacks (e.g., <0.51% recovery of phi6 used as an indicator for SARS-CoV-2 in this paper: https://www.sciencedirect.com/science/article/pii/S0048969721058009 )

My primary concern is that the PEG precipitation method does not necessarily appear to generate concentrations that provide reliable detection or trend information. For example, the authors report low concentrations in the wastewater during the Omicron surge, lack of correlation with observed clinical case trends, and lag of wastewater data (rather than lead over clinical cases). Some of these observations could be explained by poor clinical testing or other environmental factors, but it’s also possible some of the observations result from analytical issues. The analysis appears to have been performed carefully and consistently, so I wonder how effective this particular method actually is at low concentrations (which may be most relevant in these communities). Generally, we’ve found the PEG precipitation method to perform poorly. The authors should provide more context in terms of the success (or not) of the PEG method applied in generating reliable results that can be used to capture trends in infection rates for similar contexts. If other studies have demonstrated the particular utility of this method, this would build confidence in the wastewater results reported. The normalization with PMMoV also does not appear to improve the correlations with clinical cases—is this consistent with what others have seen using this method in similar contexts? Were correlations between wastewater and clinical cases in this study better or worse with and without PMMoV normalization?

We appreciate the reviewer’s concerns over some aspects of methodology. We agree that PEG is generally not as robust as other methods. Although there are some reports of higher recoveries of enveloped viruses in wastewater using PEG than other methods (e.g. of phi6 or of CHV) (Flood MT et al. Food Envrion Virol. 2021; Barril PA et al. Sci Total Environ. 2021). Other studies indicate similar performance (e.g. 44% PEG vs 28%-65% other methods) (Ahmed W et al. Sci Total Environ. 2020) or varying reductions in recovery. Given the literature supporting the efficacy of the technique, PEG precipitation does appear to be appropriate for the circumstances presented. 

The use of PPMoV for normalization does help to better analyze the data at the 3 sites. This is most apparent in the non-normalized data for Ruston which does not follow the clinical case load during the peak of Omicron in January 2022 but does correlate after normalization. This normalization is critical to the study as there are variables including rainfall that may affect concentration of virus in a given sample. The greatest challenge is likely not in the methodology, but in missed sample dates and irregular sampling intervals that result from a reliance on student and city worker schedules. Despite those missing data points, the trends in viral load correlate with reported caseloads. Finally, given the nature of wastewater testing and college towns, there is a regular fluctuation in the population that can account for discrepancies in wastewater detection, timing of detection, and correlation to caseloads. We agree that this is a valid point when assessing city wastewater in college towns and so additional text has been added to the conclusion to highlight this critical aspect of the study.

I absolutely agree that rural areas and small communities can benefit from wastewater monitoring, but more sensitive and reliable methods may be needed to confidently report results (e.g., analysis of primary clarifier sludge is much more sensitive as an overall method for detection https://pubs.rsc.org/en/content/articlehtml/2022/ew/d1ew00826a ). If the wastewater measurements don’t consistently lead over clinical cases, the tool is not useful for early warning. Only analytical results (non-normalized) from WWTP plant (Figure 2C) appears to have consistent detection for N1 and N2 as well as trends that fit with expectations from the known surges in infections. Can the authors explain the poor agreement between N1 and N2 in Figure 2a?

We agree that as a predictive tool, wastewater testing has limitations. However, as we entered a time of at-home testing, over all less testing, and in turn less reporting, wastewater-based epidemiology provides a critical accounting of virus or other detectable pathogens in a community. This is especially useful in areas where testing is limited, where there is a large population of lower income families who may not test as often for fear of missing work, and in communities where policies have been relaxed and so again, there is less testing and reporting to public health officials. The outcome of this is illustrated in Figure 2C where the wastewater data shows high levels of virus circulating in the community despite few clinical cases being reported. This coincided with relaxed policies and at-home testing where people likely had COVID-19 and were either not testing and/or not reporting to public health officials. The problem of dashboards underreporting cases and instilling a false sense of security is increasingly recognized. This points to the necessity of this type of consistent monitoring in all communities in order to best understand disease prevalence and inform public health. As pointed out, the data reported in Figure 2A does show different levels of amplification for N1 and N2. The CDC N1 primer probe set has been reported to be more sensitive than the N2 primer probe set with the N2 more likely to result in higher Cq or non-detect in a clinical setting. An assessment of 36 laboratories testing wastewater also concluded that the N1 primer probe results in higher concentrations than N2. Whether this is inherent to the primer probe sequences or due to the stability of the target regions or both is unclear but we present both sets of data. (Pecson BM et al. Environ Sci.: Water Res Technol. 2021)

I recommend that the authors report the full suite of Minimum Information for Publication of Quantitative Real-Time PCR Experiments (MIQE) to improve the completeness of the reported results ( for further information, see Bivins et al, 2021 https://www.ncbi.nlm.nih.gov/pmc/articles/PMC8341816/ ).

Some of the essential details in the MIQE checklist related to reagent/buffer compositions, target/oligonucleotide sequences can be derived from our detailed description of the materials and kits used. We have explicitly added information related to nucleic acid and qPCR QA/QC such as method blank, RNA purity, result of NTC, LOD.

Especially given higher rates of non-detects, the limit of detection (LOD) in terms of genome copies / ul extract as well as genome copies/ ml wastewater sample. Standard curves are shown, but I don’t believe the LOD was reported. How non-detects were treated in triplicate qPCR analysis should also be reported (e.g., if 1 or more results of the triplicate were below the LOD, were the replicates assigned a value such as the LOD, were the results censored, or how else where they handled?).

We have addressed these concerns in our methods section. The LOD is reported and the handling of non-detects is described.

6. PLOS authors have the option to publish the peer review history of their article (what does this mean?). If published, this will include your full peer review and any attached files.

Do you want your identity to be public for this peer review? For information about this choice, including consent withdrawal, please see our Privacy Policy.

Reviewer #1: No

Reviewer #2: No

---

## [Decision Letter · Decision Letter 1]

2 Sep 2022

Wastewater surveillance in smaller college communities may aid future public health initiatives

PONE-D-22-16566R1

Dear Dr. %Newman%,

We’re pleased to inform you that your manuscript has been judged scientifically suitable for publication and will be formally accepted for publication once it meets all outstanding technical requirements.

Kind regards,

Asli Aslan

Academic Editor

PLOS ONE

Additional Editor Comments (optional):

Reviewers' comments:

Reviewer's Responses to Questions

**Comments to the Author**

1. If the authors have adequately addressed your comments raised in a previous round of review and you feel that this manuscript is now acceptable for publication, you may indicate that here to bypass the “Comments to the Author” section, enter your conflict of interest statement in the “Confidential to Editor” section, and submit your "Accept" recommendation.

Reviewer #1: All comments have been addressed

2. Is the manuscript technically sound, and do the data support the conclusions?

Reviewer #1: Yes

3. Has the statistical analysis been performed appropriately and rigorously? 

Reviewer #1: N/A

4. Have the authors made all data underlying the findings in their manuscript fully available?

Reviewer #1: Yes

5. Is the manuscript presented in an intelligible fashion and written in standard English?

Reviewer #1: Yes

6. Review Comments to the Author

Reviewer #1: (No Response)

7. PLOS authors have the option to publish the peer review history of their article (what does this mean?). If published, this will include your full peer review and any attached files.

Reviewer #1: No

---

## [Editor Report · Acceptance letter]

7 Sep 2022

PONE-D-22-16566R1 

Wastewater surveillance in smaller college communities may aid future public health initiatives 

Dear Dr. Newman:

I'm pleased to inform you that your manuscript has been deemed suitable for publication in PLOS ONE. Congratulations! Your manuscript is now with our production department. 

Kind regards, 

on behalf of

Dr. Asli Aslan 

Academic Editor

PLOS ONE